# SPATIAL-FREQUENCY SYNERGY FOR REMOTE SENSING IMAGE SUPER-RESOLUTION WITH HOLISTIC FEATURE ENHANCEMENT

## ABSTRACT

High-resolution (HR) remote sensing images are essential for various applications of Earth observation, but hardware limitations generally give rise to low-resolution (LR) and degraded acquisitions. Super-resolution (SR) has currently emerged as a popular manner to ease this issue. However, most existing SR methods fail to effectively exploit the synergy between frequency and spatial information, while also suffering from inadequate feature enhancement. In this work, we present a novel model for remote sensing image SR, termed as Spatial-Frequency Synergy Network (SFSN). Firstly, it holistically boosts hierarchical features from both the channel and spatial dimensions, through Adaptive Channel Shifting (AdaCS) and Multi-Scale Large Kernel Attention (MS-LKA), respectively. Meanwhile, we also devise a Dual-Domain Interaction Attention (DDIA) to simulate the interaction between spatial and frequency domains explicitly, which enables synergic feature refinement and HR detail recovery. It also delivers a versatile solution for bridging the spatial-frequency domain gap in remote sensing SR. Extensive experiments on benchmark datasets have demonstrated the superiority of the proposed SFSN over advanced SR models quantitatively and qualitatively, while still maintaining considerably low overhead.

## 1 INTRODUCTION

High-resolution (HR) remote sensing images are essentially desired by a wide range of applications such as environment monitoring (Rau et al., 2014), resource exploration (Calvin et al., 2015), military reconnaissance (Wang et al., 2014) etc. However, the inherent limitations of satellite imaging hardware, transmission bandwidth, and challenging imaging environments typically lead to image degradations via resolution reduction, compression artifacts and information loss. The degradations impair critical visual features such as edge definition and fine details, substantially limiting the practical use of remote sensing images. In recent years, super-resolution (SR) has emerged as an efficient and effective alternative that can moderate the dilemma between imaging quality and overhead, which targets at recovering one HR image from its low-resolution (LR) observations.

For the task of remote sensing image SR (RSISR), deep learning methods, especially those based on convolutional neural networks (CNNs), have become a dominant solution in the field. Several representative works (Lei et al., 2017; Dong et al., 2020; Zhang et al., 2020; Lei & Shi, 2021; Wang et al., 2022) have been proposed and exhibited superior SR performance. Nevertheless, the intrinsic features of RSI, e.g., frequent recurrence of texture patterns and substantial heterogeneity of image structure, have exposed the inherent drawbacks of CNN-based approaches. The limited receptive field fails to capture long-range dependency while exhibiting insufficient adaptability to input variations, which impedes the generation of more representative features. Hence, more sophisticated models built on Transformers (Vaswani et al., 2017) have garnered increasing attention within the RSISR community (Wang et al., 2023d; Kang et al., 2024; Lei et al., 2021). Despite remarkable successes, Transformer-based methods are inclined to preserve low-frequency (LF) information while exhibiting deficient ability in recovering high-frequency (HF) details (Li et al., 2025a). Moreover, most of these SR models are computationally intensive, hindering their practical deployment in real-world applications.

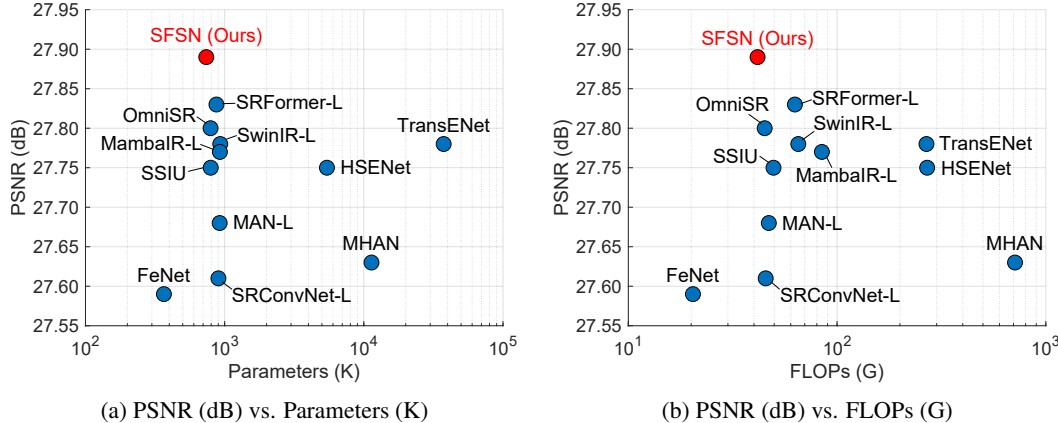

(a) PSNR (dB) vs. Parameters (K)          (b) PSNR (dB) vs. FLOPs (G)

Figure 1: Comparison between performance and overhead of typical RSISR models on UCMerced with SR×4. Our SFSN achieves the best results with the second-lowest cost.

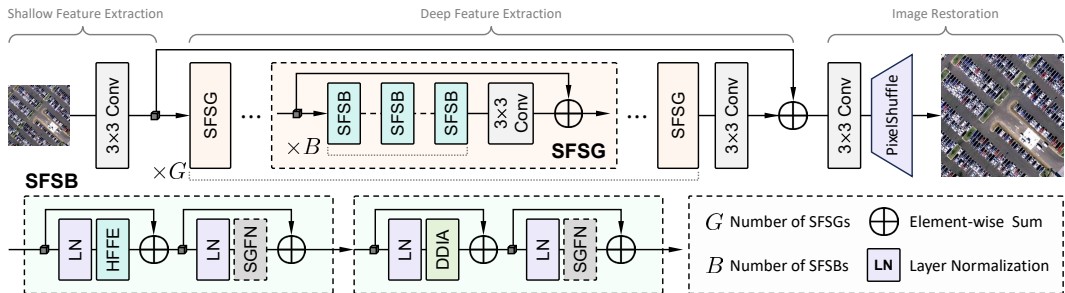

Figure 2: The overall architecture of our spatial-frequency synergy network (SFSN). Please refer to Fig. 3 and Fig. 4 for the description of our HFFE and DDIA.

Lightweight design serves as a straightforward approach to relax computational overhead, yet how to enhance model performance under constrained network resources remains an underexplored issue. Feature enhancement is a common strategy for facilitating the performance of lightweight deep models. For RSISR tasks, feature enhancement can provide more diverse features for model inference (Zhang et al., 2023) or expand the effective receptive field (ERF) of the model (Zhao et al., 2024; Li et al., 2025c), thereby improving the trade-off between model performance and cost. However, most existing approaches only consider enhancing features from either the spatial or channel dimension, failing to fully incorporate omnidirectional feature augmentation or establish effective integration strategies, which undesirably results in underexplored model capability.

Recent advances in RSISR have brought several methods (Wang et al., 2024a; Xiao et al., 2024) that fuse spatial and frequency features to enhance the capacity of SR models for accurately recovering HF structural and textural details. This dual-domain strategy can alleviate the inherent limitations of single-domain processing, i.e., *spatial features often fail to capture global information while frequency components struggle to model spatial relationships*. Nevertheless, most existing SR methods combining both spatial and frequency domains have not sufficiently explored effective strategies to harness complementary features, therefore limiting the potential for promoting model performance.

To this end, we propose a novel model in this work, which is termed as spatial-frequency synergy network (SFSN) and advances RSISR with two components: (1) holistic feature enhancement and (2) synergy inference between the spatial and frequency domains. The former adopts adaptive channel shifting (AdaCS) strategy along the channel dimension and multi-scale large-kernel attention (MS-LKA) in the spatial dimensions to enrich feature diversity and facilitate efficient model inference. The latter enables the model to reconcile long-term dependency and spatial relationship modeling via dual-domain interaction attention (DDIA). Our SFSN model achieves superior SR performance with modest parameters and costs, remarkably improving the performance-efficiency equilibrium as illustrated in Fig. 1. In summary, the main contributions of this work are as follows:

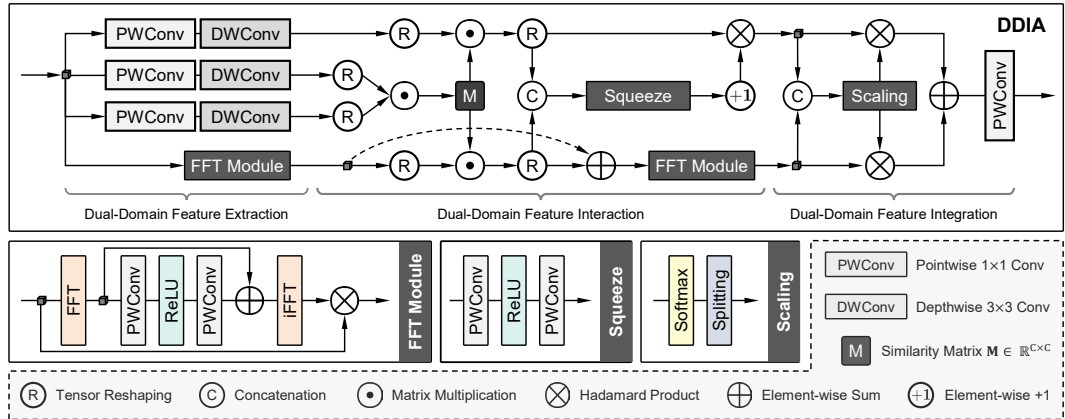

Figure 3: Structural components of our HFFE module. In AdaCS, shifting margins $(\Delta h, \Delta w)$ are determined by a Gaussian, while shifting directions are learned with a $\text{Sign}(\cdot)$ function.

Figure 4: The diagram of our DDIA. It evolves from a typical self-attention coupled with a frequency mapping branch. Please note that the operation of element-wise +1 implies a residual shortcut.

- By considering holistic feature enhancement and spatial-frequency domain interaction, we propose a lightweight yet effective SFSN for RSISR tasks. Experimental results illustrate that our SFSN achieves better compromise between model performance and overhead.

- We present a holistic fusion feature enhancement (HFFE) strategy that integrates AdaCS and MS-LKA to potentiate hierarchical features from both channel and spatial dimensions.

- A dual-domain interaction attention (DDIA) is devised to simulate the correlation between spatial and frequency domains, enabling synergic spatial-frequency feature refinement.

## 2 RELATED WORK

### 2.1 REMOTE SENSING IMAGE SUPER-RESOLUTION

Since Dong et al. (Dong et al., 2014) pioneered the usage of CNNs to single image super-resolution (SISR) and exhibited remarkable SR performance, there has been a proliferation of CNN-based methods for RSISR. Lei et al. (Lei et al., 2017) introduced LGCNet, which combines both local and global information for HR recovery. Haut et al. (Haut et al., 2019) and Dong et al. (Dong et al., 2020) incorporated residual units and skip connections to capture richer feature representation. Zhang et al. (Zhang et al., 2020) illustrated a high-order attention (HOA) module to exploit hierarchical features. Afterwards, Lei et al. (Lei & Shi, 2021) proposed to leverage multi-scale self-similarity (Glasner et al., 2009) with non-local attention. Moreover, Wang et al. (Wang et al., 2022) devised a lightweight FeNet that adopted channel splitting and weight sharing to decrease overhead.

Recently, more advanced models built upon Transformers (Vaswani et al., 2017) and Mamba (Guo et al., 2024) have garnered noteworthy attention in the RSISR community, and notably promoted the progress of RSISR. The representative works include TransNet (Lei et al., 2021), HAUNet (Wang et al., 2023d), ESTNet (Kang et al., 2024), as well as FMSR (Xiao et al., 2024) and ConvMambaSRF (Zhu et al., 2024). Although these models have made notable progresses in modeling long-range dependencies, they typically ignore the benefits of adequate feature enhancement for RSISR.

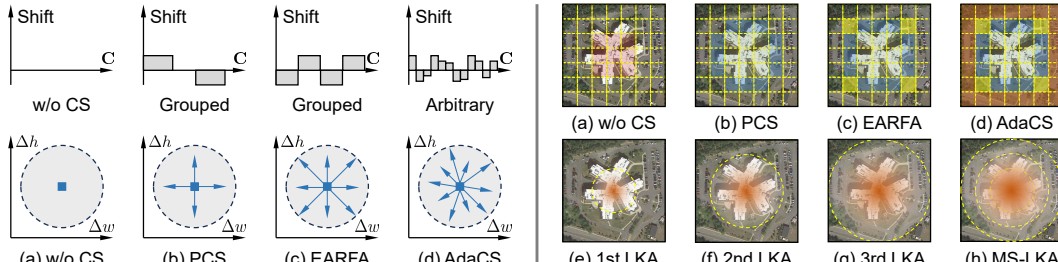

Figure 5: The diagram of our AdaCS (left) and the synergy with MS-LKA (right). Different from PCS (Zhang et al., 2023) and EARFA (Zhao et al., 2024), our AdaCS conducts channel shifting on non-grouped channels with arbitrary amplitude and direction. In collaboration with MS-LKA, our HFFE achieves holistic feature enhancement ((d) + (h) on the right side).

Furthermore, when the model scale is limited, the representational capability of the model has not been fully exploited, resulting in suboptimal trade-offs between model performance and overhead.

## 2.2 FEATURE ENHANCEMENT FOR SISR

Feature enhancement serves as a prevalent tool to augment the representational capacity of SR models by promoting the effectiveness of feature representation. One common type of feature enhancement approaches for SR tasks is attention mechanism, including channel attention (Hu et al., 2018), spatial attention (Hu et al., 2019), and channel-spatial attention (Niu et al., 2020), as well as prior-guided non-local attention (Mei et al., 2023) and large kernel attention (Guo et al., 2023a; Li et al., 2025c) etc. Another more direct method involves partially shifting feature channels through primitive data movement with negligible cost. For instance, Zhang et al. (Zhang et al., 2023) proposed the PCS strategy to shift partial channels, which can intuitively enlarge the ERF of the model. Zhao et al. (Zhao et al., 2024) introduced the SLKA module, integrating channel shifting and large kernel attention to further expand the ERF. However, these models only performed fixed-direction and fixed-amplitude shifts on grouped channels, failing to thoroughly exploit the potential of channel shifting. Therefore, we designed a HFFE module for more comprehensive feature enhancement with channel-wise shifting and spatial MS-LKA.

## 2.3 FOURIER-DOMAIN LEARNING FOR IMAGE RESTORATION

As a fundamental tool for spectral analysis, the fast Fourier transform (FFT) provides unique benefits in modeling global signal dependencies, leading to its broad adoption in various visual tasks. Mao et al. (Mao et al., 2021) devised a ResFFT module that can integrate LF and HF residual information simultaneously. Guo et al. (Guo et al., 2023b) proposed a window-based frequency channel attention that utilizes FFT to exploit richer global information. Similarly, Chen et al. (Chen et al., 2023a) combined Swin Transformer layers with fast Fourier convolution (FFC) to demodulate both local and global features. Wang et al. (Wang et al., 2023a) conducted mutual learning between frequency and spatial domains to boost the performance of face super-resolution (FSR). And Wang et al. (Wang et al., 2024a) developed a TSFNet model to combine spatial and frequency information from coarse to fine, which progressively refined the result of RSISR. These studies illustrate the efficacy of Fourier-domain learning in image restoration. However, current practice for integrating spatial-frequency features mainly resort to simple operations like concat or element-wise addition, failing to fully explore the synergy between the spatial and Fourier domains.

## 3 METHODOLOGY

### 3.1 OVERALL ARCHITECTURE

The overall structure of our SFSN is demonstrated in Fig. 2, which follows the typical paradigm consisting of shallow feature extraction, deep feature extraction, as well as image restoration. Let $\mathbf{x} \in \mathbb{R}^{H \times W \times 3}$ represent the input, and $\mathbf{y} \in \mathbb{R}^{rH \times rW \times 3}$ to be the output of the model, where $H$ and

Table 1: Details of the datasets. "Res." stands for the spatial resolution of pixels (m/pixel).

| Datasets | Division | | | Size | Classes | Res. |
|---|---|---|---|---|---|---|
| | Train | Valid | Test | | | |
| UCMerced | 945 | 105 | 1050 | $256^2$ | 21 | 0.3 |
| RSSCN7 | 1120 | 280 | 1400 | $400^2$ | 7 | / |
| AID | 7850 | 150 | 2000 | $600^2$ | 30 | 0.5 |

Table 2: Ablation study of the proposed SFSB on UCMerced with SR×2.

| Variant | Params [K] | FLOPs [G] | UCMerced | |
|---|---|---|---|---|
| | | | PSNR | SSIM |
| HFFE + HFFE | 692 | 153.4 | 34.44 | 0.9341 |
| DDIA + DDIA | 754 | 172.6 | 34.51 | 0.9330 |
| DDIA + HFFE | 723 | 163.0 | 34.49 | 0.9345 |
| HFFE + DDIA | 723 | 163.0 | 34.57 | 0.9350 |

$W$ denote the height and width of the input image, and $r$ stands for the scaling factor. The shallow feature $\mathbf{x}_0 \in \mathbb{R}^{H \times W \times C}$ is captured with a single 3×3 conv layer, where $C$ denotes the number of feature channels. Then, the deep feature $\mathbf{x}_G \in \mathbb{R}^{H \times W \times C}$ can be generated via a stack of spatial frequency synergy groups (SFSG), where $G$ represents the number of SFSG. The model yields HR image $\mathbf{y}$ as the following:

$$\mathbf{y} = \text{PixelShuffle}\big(\text{Conv}_{3 \times 3}\big(\mathbf{x}_0 + \text{Conv}_{3 \times 3}\big(\mathbf{x}_G\big)\big)\big), \tag{1}$$

where PixelShuffle$(\cdot)$ denotes the upscaling layer to enlarge the final feature to the target resolution, and Conv$_{3 \times 3}(\cdot)$ is 3×3 conv layers. $\mathbf{y}$ is utilized to construct $L_1$ loss for model training. Within each SFSG, the inference follows a similar procedure, where one SFSG includes $B$ spatial-frequency synergy blocks (SFSBs) followed by a 3×3 conv layer and a residual shortcut.

The core components of our SFSN model, i.e., HFFE and DDIA, are contained within the SFSB. Together with SGFN (Chen et al., 2023b), they constitute two building units akin to the Transformer, as shown in Fig. 2. Suppose the input of the first unit is $\mathbf{x}_t$ and the output is $\mathbf{z}_t$, then the mapping can be formulated as ($t$ means "temporary"):

$$\mathbf{z}_t = \mathcal{S}(\text{LN}(\mathbf{y}_t)) + \mathbf{y}_t, \quad \mathbf{y}_t = \mathcal{H}(\text{LN}(\mathbf{x}_t)) + \mathbf{x}_t, \tag{2}$$

where $\mathbf{y}_t$ implies an intermediate feature, and LN$(\cdot)$ denotes layer normalization. $\mathcal{H}(\cdot)$ and $\mathcal{S}(\cdot)$ represent the mappings of HFFE and SGFN (Chen et al., 2023b), respectively. Similarly, the second unit with DDIA can be written as:

$$\mathbf{z}_t = \mathcal{S}(\text{LN}(\mathbf{y}_t)) + \mathbf{y}_t, \quad \mathbf{y}_t = \mathcal{D}(\text{LN}(\mathbf{x}_t)) + \mathbf{x}_t, \tag{3}$$

where $\mathcal{D}(\cdot)$ stands for the function of DDIA.

## 3.2 Holistic Fusion Feature Enhancement

The HFFE is designed to comprehensively enhance features from both channel and spatial dimensions, where channel-wise enhancement is achieved through AdaCS, and spatial enhancement is in virtue of MS-LKA. The detailed structure of HFFE is illustrated in Fig. 3.

**AdaCS**: Given a temporary feature $\mathbf{x}_t \in \mathbb{R}^{H \times W \times C}$, AdaCS is devised to learn two shifting amplitudes, $\Delta h$ and $\Delta w$, for each channel of $\mathbf{x}_t$. The structure of our AdaCS is presented in Fig. 3, where $\mathbf{s} \in \mathbb{R}^{C \times 2}$ stands for the shifting vector containing all amplitudes of $C$ channels. The shifting vector $\mathbf{s}$ consists of two parts: (1) the shifting directions are learned by the network using a Sign$(\cdot)$ function; and (2) the shifting magnitudes are randomly assigned via a Gaussian $\mathcal{N}(\mu, \sigma)$. The procedure of obtaining $\mathbf{s}$ can be described as:

$$\mathbf{y}_t = \text{GAP}(\mathbf{x}_t), \quad \mathbf{z}_t = \text{PWConv}(\text{ReLU}(\text{PWConv}(\mathbf{y}_t))), \quad \mathbf{s} = \text{Sign}(\mathbf{z}_t) \otimes \mathcal{N}(\mu, \sigma), \tag{4}$$

where PWConv$(\cdot)$ denotes a 1×1 point-wise conv layer, and GAP$(\cdot)$ implies global average pooling. $\mu$ and $\sigma$ indicate the mean and standard deviation of the Gaussian. We implement channel-wise shifting on $\mathbf{x}_t$ via bilinear interpolation to deal with data movement in non-integer grids. The final output of our AdaCS can be expressed as:

$$\text{AdaCS}(\mathbf{x}_t) = \text{PWConv}(\text{CS}(\mathbf{x}_t, \mathbf{s})), \tag{5}$$

where CS$(\mathbf{x}_t, \mathbf{s})$ means to shift $\mathbf{x}_t$ with the shifting vector $\mathbf{s}$ using bilinear interpolation. It is noteworthy that restricting the model to learn only the shifting directions helps reduce training complexity, while obtaining the shifting magnitudes with Gaussian sampling provides operational flexibility that permits explicit control over the shifting range. Thereby, our AdaCS enables arbitrary-magnitude shifts for non-grouped channels, significantly different from existing approaches, as shown in Fig. 5.

Table 3: Ablation study of our HFFE on UCMerced with SR×2. For convenience, we use $\mathbf{B}_1$, $\mathbf{B}_2$ and $\mathbf{B}_3$ to represent the 1st, 2nd and 3rd LKA branches (refer to Fig. 3), respectively. Please note that if only one LKA branch is deployed, then there is no channel splitting.

| Metrics | Full HFFE | w/o HFFE | w/o AdaCS | AdaCS $\rightarrow$ CS | LKA Branches | | | | | |
|---|---|---|---|---|---|---|---|---|---|---|
| | | | | | $\mathbf{B}_1$ | $\mathbf{B}_2$ | $\mathbf{B}_3$ | $\mathbf{B}_1 + \mathbf{B}_2$ | $\mathbf{B}_1 + \mathbf{B}_3$ | $\mathbf{B}_2 + \mathbf{B}_3$ |
| Params [K] | 723 | 629 | 693 | 718 | 717 | 728 | 724 | 723 | 721 | 726 |
| FLOPs [G] | 163.0 | 143.5 | 158.2 | 162.9 | 161.7 | 164.1 | 163.3 | 162.9 | 162.5 | 163.7 |
| PSNR [dB] | 34.57 | 34.39 | 34.49 | 34.52 | 34.53 | 34.51 | 34.52 | 34.54 | 34.55 | 34.53 |
| SSIM | 0.9350 | 0.9335 | 0.9342 | 0.9347 | 0.9347 | 0.9344 | 0.9346 | 0.9346 | 0.9348 | 0.9341 |

**MS-LKA**: LSKNet (Li et al., 2023) conducted a large kernel selection mechanism to enhance features through leveraging the prior of RSI that the contextual information for different objects is very different. We take a further step for this from a multi-scale perspective, as shown in Fig. 3. For a feature $\mathbf{x}_t$ that has freshly undergone the process of our AdaCS, it is evenly divided into three sub-features $\mathbf{x}_t^1$, $\mathbf{x}_t^2$, and $\mathbf{x}_t^3$ along the channel dimension:

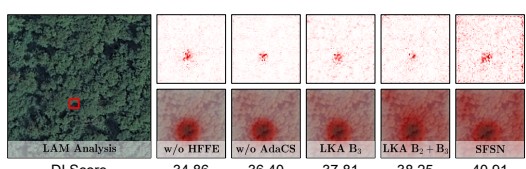

Figure 6: LAM results of different variants of our SFSN on "forest76.tif" from UCMerced.

$$[\mathbf{x}_t^1, \mathbf{x}_t^2, \mathbf{x}_t^3] = \text{Split}(\text{AdaCS}(\mathbf{x}_t)). \tag{6}$$

Each sub-feature is then enhanced by a LKA branch: $\hat{\mathbf{x}}_t^i = \text{LKA}(\mathbf{x}_t^i)$, $i = 1, 2, 3$. We employ two decomposed depth-wise convolutions (DWConv) with different kernel sizes or dilation rates to extract features within each LKA:

$$\mathbf{m}_1^i = \text{DWConv}(\mathbf{x}_t^i), \quad \mathbf{m}_2^i = \text{DWConv}(\mathbf{m}_1^i). \tag{7}$$

Let $(k_1, d_1)$ and $(k_2, d_2)$ denote the kernel size and dilation rate of these two layers, respectively. In our implementation, the 1st LKA branch keeps $(k_1, d_1) = (3, 3)$ and $(k_2, d_2) = (5, 3)$. For the 2nd LKA branch, $(k_1, d_1) = (3, 1)$ and $(k_2, d_2) = (7, 3)$. We set $(k_1, d_1) = (5, 1)$ and $(k_2, d_2) = (5, 5)$ for the last branch. We obtain the composite spatial attention map via $\mathbf{m}^i = \text{Concat}(\mathbf{m}_1^i, \mathbf{m}_2^i)$. The weights for selective LKA are generated by applying a PWConv and a Sigmoid activation:

$$[\mathbf{w}_1^i, \mathbf{w}_2^i] = \text{Split}(\text{Sigmoid}(\text{PWConv}(\mathbf{m}^i))), \tag{8}$$

where $\text{Split}(\cdot)$ divides these weights into two parts along the channels. The output of the $i$-th LKA branch is obtained via:

$$\hat{\mathbf{x}}_t^i = \text{LKA}(\mathbf{x}_t^i) = \mathbf{m}_1^i \otimes \mathbf{w}_1^i + \mathbf{m}_2^i \otimes \mathbf{w}_2^i. \tag{9}$$

The inference of the above LKA is akin to LSKNet (Li et al., 2023), but works in a simpler and more effective way. Finally, our MS-LKA yields the following:

$$\hat{\mathbf{x}}_t = \text{GeLU}(\text{PWConv}(\text{Concat}(\hat{\mathbf{x}}_t^1, \hat{\mathbf{x}}_t^2, \hat{\mathbf{x}}_t^3))). \tag{10}$$

The enhanced feature generated by our HFFE is obtained via $\mathcal{H}(\mathbf{x}_t) = \hat{\mathbf{x}}_t \otimes \mathbf{x}_t$, which can holistically promote feature effectiveness in virtue of our AdaCS and MS-LKA.

## 3.3 DUAL-DOMAIN INTERACTION ATTENTION

The DDIA is motivated by the synergism between spatial and frequency domains, as illustrated in Fig. 4. It consists of a spatial branch that adopts a self-attention-like structure to extract features, and a FFT branch that captures frequency information with a FFT module.

**Dual-Domain Feature Extraction**: Given a temporary feature $\mathbf{x}_t$, we employ a self-attention-like structure to extract three spatial features $\mathbf{K}_t, \mathbf{Q}_t, \mathbf{V}_t \in \mathbb{R}^{H \times W \times C}$ using a PWConv followed by a DWConv, as shown in Fig. 4. The Fourier feature $\mathbf{F}_t$ is captured by a FFT module: $\mathbf{F}_t = \mathcal{F}(\mathbf{x}_t) \in \mathbb{R}^{H \times W \times C}$, where $\mathcal{F}(\cdot)$ denotes the mapping of the module. Let $\mathbf{y}_t = \text{FFT}(\mathbf{x}_t)$ to be the Fourier transform of $\mathbf{x}_t$, then the procedure of generating $\mathbf{F}_t$ can be formulated as:

$$\mathbf{z}_t = \text{PWConv}(\text{ReLU}(\text{PWConv}(\mathbf{y}_t))), \quad \mathbf{F}_t = \mathbf{x}_t \otimes \text{iFFT}(\mathbf{y}_t + \mathbf{z}_t), \tag{11}$$

where $\text{iFFT}(\cdot)$ is the inverse FFT. This process is depicted in the FFT module of Fig. 4.

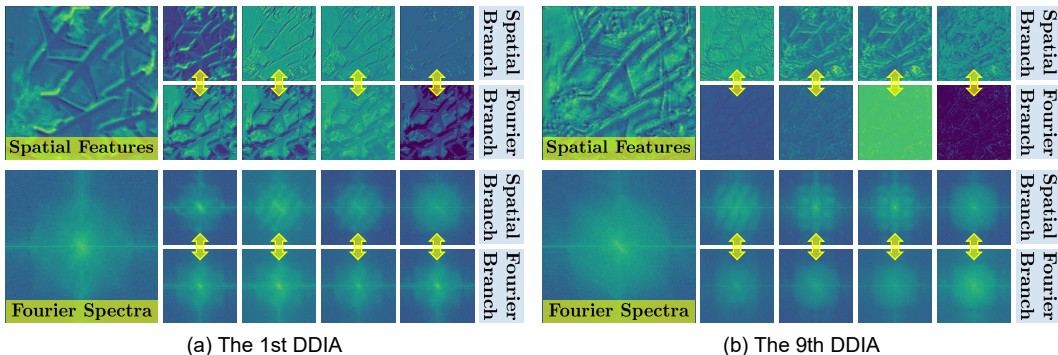

Spatial Features | Fourier Spectra | (a) The 1st DDIA

Spatial Features | Fourier Spectra | (b) The 9th DDIA

Figure 7: The visualization of spatial features and Fourier spectra for the 1st and 9th DDIA modules of our SFSN model. The testing image is "airplane87.tif" from UCMerced (Yang & Newsam, 2010).

**Dual-Domain Feature Interaction**: All features mentioned above are reshaped into $\mathbb{R}^{HW \times C}$ for feature interaction. We first obtain a non-negative similarity matrix $\mathbf{M} \in \mathbb{R}^{C \times C}$ through:

$$\mathbf{M} = \text{ReLU}\big[\mathcal{R}(\mathbf{Q}_t)^T \cdot \mathcal{R}(\mathbf{K}_t)/\sqrt{d_K}\big], \tag{12}$$

where $d_K$ is a learnable parameter that is initialized as $C$, and $\mathcal{R}(\cdot)$ implies an operation of tensor reshaping. Next, we fuse spatial and spectral features as following:

$$\hat{\mathbf{V}}_t = \mathcal{R}(\mathcal{R}(\mathbf{V}_t) \cdot \mathbf{M}), \quad \hat{\mathbf{F}}_t = \mathcal{R}(\mathcal{R}(\mathbf{F}_t) \cdot \mathbf{M}). \tag{13}$$

Then we get a new feature for the spatial branch:

$$\tilde{\mathbf{V}}_t = \hat{\mathbf{V}}_t \otimes \big[\text{Squeeze}\big(\text{Concat}(\hat{\mathbf{V}}_t, \hat{\mathbf{F}}_t)\big) + 1\big] = \hat{\mathbf{V}}_t \otimes \text{Squeeze}\big(\text{Concat}(\hat{\mathbf{V}}_t, \hat{\mathbf{F}}_t)\big) + \hat{\mathbf{V}}_t, \tag{14}$$

where $\text{Squeeze}(\cdot)$ is composed of two PWConv layers, with a ReLU activation sandwiched between them. Please note that +1 implicitly indicates a residual shortcut. Similarly, a new spectral feature is produced by $\tilde{\mathbf{F}}_t = \mathcal{F}(\hat{\mathbf{F}}_t + \mathbf{F}_t)$.

**Dual-Domain Feature Integration**: Subsequently, we fuse $\tilde{\mathbf{V}}_t$ and $\tilde{\mathbf{F}}_t$ through a simple feature scaling. We first obtain two weights for feature integration as following:

$$\mathbf{w}_t^v = e^{\tilde{\mathbf{V}}_t}/(e^{\tilde{\mathbf{V}}_t} + e^{\tilde{\mathbf{F}}_t}), \qquad \mathbf{w}_t^f = e^{\tilde{\mathbf{F}}_t}/(e^{\tilde{\mathbf{V}}_t} + e^{\tilde{\mathbf{F}}_t}). \tag{15}$$

This is conducted by concatenating $\tilde{\mathbf{V}}_t$ and $\tilde{\mathbf{F}}_t$ along a new dimension and feeding them into a softmax function. So the outputs of feature integration, which is also the final output of our DDIA $\mathcal{D}(\mathbf{x}_t)$, can be expressed as: $\mathcal{D}(\mathbf{x}_t) = \text{PWConv}\big(\mathbf{w}_t^v \cdot \tilde{\mathbf{V}}_t + \mathbf{w}_t^f \cdot \tilde{\mathbf{F}}_t\big)$.

## 4 EXPERIMENT

### 4.1 DATASETS AND METRICS

We employ three public datasets for evaluation, i.e., AID (Xia et al., 2017), RSSCN7 (Zou et al., 2015), and UCMerced (Yang & Newsam, 2010), and the detailed information for them is shown in Table 1. We evaluate the quantitative performance of the compared approaches through Peak Signal-to-Noise Ratio (PSNR) and Structural Similarity (SSIM) (Wang et al., 2004).

### 4.2 IMPLEMENTATION DETAILS

The numbers of SFSGs, SFSBs and feature channels are set to 3, 3, and 48 respectively, and we use 6 heads for self-attention. The ratio of channel expansion for the SGFN (Chen et al., 2023b) is set to 6. All versions of our SFSN are trained with a LR patch size of 64×64 (HR patch size relies on scaling factors) for 500K iterations using a batch size of 8. We train our models using the Adam optimizer (Adam et al., 2014) with its default parameter settings. All models are implemented with PyTorch and trained on a workstation with 4 NVIDIA GeForce RTX A5000 GPUs. Besides, the FLOPs of the models are obtained based on HR images with a spatial resolution of 1280×720 pixels.

## 4.3 ABLATION STUDY

In this section, we validate the effectiveness of the proposed components from coarse to fine. For fair comparisons, we implement all ablation experiments using the same settings as the proposed SFSN, using UCMerced dataset with SR×2 as the testing dataset.

**Components of SFSB**: We compare three variants of SFSB regarding the combination of HFFE and DDIA, as exhibited in Table 2. It can be seen that even though SFSB results in moderate parameters and FLOPs, it illustrates the best SR performance. Moreover, comparing rows #3 and #4 reveals that performing feature enhancement prior to dual-domain feature interaction benefits model inference.

**Effectiveness of HFFE**: To demonstrate the effectiveness of our HFFE, we simply remove it

Table 4: Ablation study on our DDIA with SR×2. **A**: Self-Attention; **B**: FFT Branch; **C**: Scaling; **D**: Interaction; **E**: Squeeze.

| A | B | C | D | E | Params [K] | FLOPs [G] | UCMerced PSNR | SSIM |
|---|---|---|---|---|---|---|---|---|
| ✗ | ✗ | ✗ | ✗ | ✗ | 598 | 133.9 | 34.35 | 0.9332 |
| ✓ | ✗ | ✗ | ✗ | ✗ | 693 | 155.7 | 34.40 | 0.9335 |
| ✓ | ✓ | ✗ | ✗ | ✗ | 715 | 160.7 | 34.45 | 0.9344 |
| ✓ | ✓ | ✓ | ✗ | ✗ | 715 | 161.2 | 34.47 | 0.9347 |
| ✓ | ✓ | ✓ | ✓ | ✗ | 715 | 161.8 | 34.53 | 0.9348 |
| ✓ | ✓ | ✓ | ✓ | ✓ | 723 | 163.0 | 34.57 | 0.9350 |

from SFSB and this results in a PSNR degradation of 0.18dB, as shown in Table 3. To inspect the effect of our AdaCS, we conduct experiments by removing it or replacing it with CS (Zhao et al., 2024), which leads to the results of 34.49 dB and 34.52 dB. Moreover, we implement six variants for MS-LKA: the first three variants for a single scale and the last three for two scales. The results in Table 3 prove that progressively increasing scale diversity generally leads to performance gains.

**Effectiveness of DDIA**: The DDIA module can be regarded as being composed of 5 components: Self-attention, FFT branch, Scaling, Interaction and Squeeze, as shown in Table 4 and Fig.4. The baseline without any component (i.e., without DDIA) achieves a PSNR value of 34.35 dB. Deploying SA on this basic model results in gains of 0.05 dB and further integration of FFT branch delivers an extra increase of 0.05 dB. Other components also contribute to the performance of the model to some extent. The results collectively validate the efficacy of the proposed DDIA.

## 4.4 LAM ANALYSIS AND FEATURE VISUALIZATION

**LAM Analysis on HFFE**: We leverage local attribution map (LAM) (Gu & Dong, 2021) to demonstrate the contribution of our HFFE to feature enhancement, which quantifies the impact of local regions in LR images on SR results. Fig. 6 clarifies that our HFFE and its components significantly contribute to the perception of the model to the surrounding information, which can also be verified through the diffusion indexes (DI) below each comparative group.

**Feature Visualization of DDIA**: RSI images often exhibit intricate texture distributions and varying scales of objects (Wang et al., 2023c), yet they also involve fewer extremely HF details commonly found in natural images, which means that the main features of RSI images may be distributed in certain spectral bands.

We visualize the internal features of our DDIA in the first and last SFSBs, as shown in Fig. 7. For each comparison group, the 1st and 2nd rows represent the spatial and Fourier branches, respectively. The 2×4 small images correspond to the average feature maps of $\mathbf{V}_t/\mathbf{F}_t$, $\hat{\mathbf{V}}_t/\hat{\mathbf{F}}_t$, $\tilde{\mathbf{V}}_t/\tilde{\mathbf{F}}_t$, and $\mathbf{w}_t^v \cdot \tilde{\mathbf{V}}_t/\mathbf{w}_t^f \cdot \tilde{\mathbf{F}}_t$, from left to right. The large image in each comparison group is the input feature or its Fourier spectrum of each DDIA. It can be seen that the Fourier branch helps to modulate the features of the spatial branch in some specific spectral bands, allowing the model to gradually learn more refined structural and spectral details.

## 4.5 COMPARISON WITH ADVANCED METHODS

We compare our SFSN with several advanced SR methods including: SwinIR (Liang et al., 2021), TransENet (Lei et al., 2021), FeNet (Wang et al., 2022), OmniSR (Wang et al., 2023b), SRFormer (Zhou et al., 2023), MAN (Wang et al., 2024b), SRConvNet (Li et al., 2025b), MambaIR (Guo et al., 2024), SSIU (Ni et al., 2025).

**Quantitative Comparison**: Table 5 shows the quantitative results of compared approaches. As can be seen, our SFSN presents the best performance across all benchmark datasets for various

Table 5: Quantitative results of advanced SISR models. SFSN+ represents the enhanced version of our model with geometric self-ensemble (Lim et al., 2017). The best and second-best results are marked in red and blue, respectively.

| Scales | SISR Models | Annual | Params [K] | FLOPs [G] | UCMerced | | RSSCN7 | | AID | |
|---|---|---|---|---|---|---|---|---|---|---|
| | | | | | PSNR | SSIM | PSNR | SSIM | PSNR | SSIM |
| SR×2 | SwinIR-L | ICCV2021 | 910 | 252.9 | 34.48 | 0.9343 | 30.18 | 0.8082 | 35.46 | 0.9378 |
| | TransENet | TGRS2022 | 37311 | 550.5 | 34.05 | 0.9294 | 30.08 | 0.8040 | 35.40 | 0.9372 |
| | FeNet | TGRS2022 | 351 | 77.9 | 33.95 | 0.9284 | 30.05 | 0.8033 | 35.33 | 0.9364 |
| | OmniSR | CVPR2023 | 772 | 172.1 | 34.16 | 0.9303 | 30.11 | 0.8052 | 35.50 | 0.9383 |
| | SRFormer-L | ICCV2023 | 853 | 236.2 | 34.53 | 0.9347 | 30.23 | 0.8101 | 35.52 | 0.9384 |
| | MAN-L | CVPR2024 | 823 | 184.0 | 34.44 | 0.9341 | 30.18 | 0.8093 | 35.53 | 0.9389 |
| | MambaIR-L | ECCV2024 | 905 | 334.2 | 34.45 | 0.9334 | 30.22 | 0.8098 | 35.51 | 0.9381 |
| | SRConvNet-L | IJCV2025 | 885 | 160.1 | 34.35 | 0.9333 | 30.20 | 0.8090 | 35.50 | 0.9381 |
| | SSIU | TIP2025 | 778 | 164.5 | 34.50 | 0.9344 | 30.22 | 0.8094 | 35.52 | 0.9384 |
| | SFSN (Ours) | — | 723 | 163.0 | 34.57 | 0.9350 | 30.26 | 0.8113 | 35.56 | 0.9397 |
| | SFSN+ (Ours) | — | 723 | 163.0 | 34.75 | 0.9365 | 30.30 | 0.8123 | 35.67 | 0.9401 |
| SR×3 | SwinIR-L | ICCV2021 | 918 | 114.5 | 30.15 | 0.8466 | 28.05 | 0.7081 | 31.53 | 0.8596 |
| | TransENet | TGRS2022 | 37496 | 357.5 | 29.90 | 0.8397 | 28.02 | 0.7054 | 31.50 | 0.8588 |
| | FeNet | TGRS2022 | 357 | 35.2 | 29.80 | 0.8379 | 27.97 | 0.7031 | 31.33 | 0.8550 |
| | OmniSR | CVPR2023 | 780 | 78.0 | 29.99 | 0.8403 | 28.04 | 0.7061 | 31.53 | 0.8596 |
| | SRFormer-L | ICCV2023 | 861 | 104.8 | 30.23 | 0.8502 | 28.07 | 0.7089 | 31.57 | 0.8607 |
| | MAN-L | CVPR2024 | 832 | 81.3 | 30.08 | 0.8446 | 28.02 | 0.7072 | 31.53 | 0.8607 |
| | MambaIR-L | ECCV2024 | 913 | 148.5 | 30.15 | 0.8468 | 28.08 | 0.7097 | 31.56 | 0.8601 |
| | SRConvNet-L | IJCV2025 | 906 | 74.8 | 29.89 | 0.8382 | 28.02 | 0.7063 | 31.53 | 0.8595 |
| | SSIU | TIP2025 | 799 | 75.1 | 30.17 | 0.8499 | 28.06 | 0.7082 | 31.57 | 0.8609 |
| | SFSN (Ours) | — | 729 | 71.9 | 30.29 | 0.8519 | 28.10 | 0.7102 | 31.60 | 0.8622 |
| | SFSN+ (Ours) | — | 729 | 71.9 | 30.44 | 0.8550 | 28.14 | 0.7115 | 31.67 | 0.8628 |
| SR×4 | SwinIR-L | ICCV2021 | 930 | 65.2 | 27.78 | 0.7662 | 26.83 | 0.6391 | 29.35 | 0.7883 |
| | TransENet | TGRS2022 | 37459 | 268.0 | 27.78 | 0.7635 | 26.81 | 0.6372 | 29.44 | 0.7912 |
| | FeNet | TGRS2022 | 366 | 20.4 | 27.59 | 0.7538 | 26.80 | 0.6367 | 29.16 | 0.7812 |
| | OmniSR | CVPR2023 | 792 | 45.0 | 27.80 | 0.7637 | 26.85 | 0.6388 | 29.19 | 0.7829 |
| | SRFormer-L | ICCV2023 | 873 | 62.8 | 27.83 | 0.7680 | 26.84 | 0.6400 | 29.39 | 0.7895 |
| | MAN-L | CVPR2024 | 921 | 47.1 | 27.68 | 0.7638 | 26.77 | 0.6382 | 29.35 | 0.7891 |
| | MambaIR-L | ECCV2024 | 924 | 84.6 | 27.77 | 0.7666 | 26.84 | 0.6400 | 29.37 | 0.7882 |
| | SRConvNet-L | IJCV2025 | 902 | 45.5 | 27.61 | 0.7618 | 26.79 | 0.6366 | 29.33 | 0.7873 |
| | SSIU | TIP2025 | 794 | 49.6 | 27.75 | 0.7652 | 26.83 | 0.6392 | 29.40 | 0.7894 |
| | SFSN (Ours) | — | 738 | 41.6 | 27.89 | 0.7699 | 26.87 | 0.6419 | 29.46 | 0.7913 |
| | SFSN+ (Ours) | — | 738 | 41.6 | 28.09 | 0.7747 | 26.92 | 0.6437 | 29.51 | 0.7923 |

scale factors. Notably, it achieves the best results with second-lowest model parameters and FLOPs, striking a better trade-off between model performance and overhead. A similar conclusion can also be drawn from Fig. 1.

**Qualitative Comparison**: We present the visual comparison of these methods in Fig. 8, where two images from AID and UCMerced are used for evaluation with SR×4. It can be observed that our SFSN model is capable of reconstructing the latent contours of the objects more reliably, which poses a significant impact on recognition, detection and counting tasks. However, most other methods fail to correctly recover this scenario. For instance, the results of some SR methods erroneously blend the dock and small boats together, such as SwinIR, FeNet, SRFormer and SSIU.

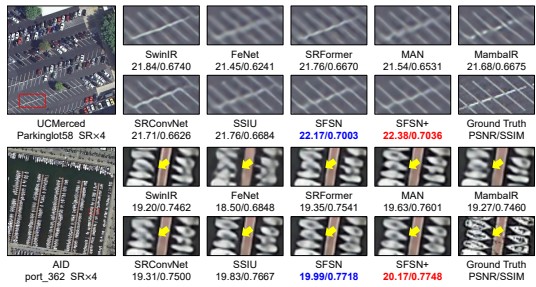

Figure 8: Visual results of compared RSISR models with SR×4. The best and second-best results are marked in red and blue, respectively.

## 5 CONCLUSION

In this work, we propose a lightweight SFSN for the RSISR task, which comprehensively lifts model capability through the perspectives of feature enhancement and the synergism between spatial and frequency domains. We leverage AdaCS and MS-LKA operating on channel and spatial dimensions respectively to comprehensively enhance features, while the spatial-frequency domain synergism is implemented via an attention mechanism assisted by a Fourier branch. Extensive experiments demonstrate that, with the aid of holistic feature enhancement, the proposed SFSN model effectively alleviates the problem of long-range dependencies and spatial relationship modeling confronted by single domain processing in RSISR tasks, and it also strikes a better compromise between model performance and complexity.

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

# A APPENDIX

## A.1 INTRODUCTION

We include some supplementary explanations with regard to our SFSN model, implementation details, and experimental comparisons in this document:

- **Channel Splitting**: Given the shifting vector $\mathbf{s}$, we detail how to shift the channels of intermediate features, which was omitted in the main text due to page limits.

- **Implementation Details**: We provide an explanation on the implementation of the LKA of our HFFE.

- **Ablation Study**: Due to the complex architecture of our DDIA, we elaborately show the structures of the variants in its ablation experiments.

- **Experiments**: We also add some additional experimental results here, including LAM analysis, quantitative results with other models, and extra visual comparisons.

## A.2 ADDITIONAL IMPLEMENTATION DETAILS

## A.3 CHANNEL SHIFTING

Within our AdaCS, we utilize $\mathrm{CS}(\mathbf{x}_t, \mathbf{s})$ to shift intermediate feature $\mathbf{x}_t$ through the shifting vector $\mathbf{s}$. The implementation details of $\mathrm{CS}(\mathbf{x}_t, \mathbf{s})$ are demonstrated in Algorithm 1. It first generates shifted coordinate grids with $\mathbf{s}$, and then performs feature shifting using bilinear interpolation.

The source code and pre-trained models of our SFSN will be publicly released on Github.

## A.4 EXPERIMENTS

### A.4.1 ABLATION STUDY

**Explanation of Variants of MS-LKA**: In the ablation study of MS-LKA, we have six variants: the first three variants use one single LKA without channel splitting (48 channels for mapping), and the last three ones employ two different LKAs, each of which possesses a half of the channels of the original feature (24 out of 48).

**Explanation of Variants of DDIA**: It can be seen in Fig. 9 that our DDIA consists of five components: Self-attention (**A**), FFT branch (**B**), Scaling (**C**), Interaction (**D**), Squeeze (**E**). In Fig. 9, the variant **A** indicates that only component **A** is used, and the variant **B** incorporates component **B** based on variant **A**. Similarly, the variant **C** is built through adding component **C** to the preceding variant **B**, and the variant **D** is formed by adding component **D** to the variant **C**, as shown in Fig. 10. Besides, the entire DDIA module is constructed by supplementing component **E** to the variant **D**.

### A.4.2 COMPARISON EXPERIMENTS

**LAM Comparison with SRConvNet**: To better exhibit the performance advantages of our SFSN, we leverage LAM to conduct attribution analyses on both SFSN and the advanced SRConvNet (Li et al., 2025b). Through this side-by-side comparison, we can intuitively contrast the critical visual cues relied upon by both models during their decision-making processes, thus enabling a more comprehensive evaluation of their performance. As shown in Fig. 11, it is evident that our model is capable of reconstructing target regions through larger spatial areas. This indicates that it can leverage texture information from a broader scope than SRConvNet (Li et al., 2025b) to facilitate more accurate detail reconstruction.

**More Quantitative Results**: Due to the page constraint, we omitted quantitative comparisons with several representative SR models in the main document, including SRCNN (Dong et al., 2014), MHAN (Zhang et al., 2020), HSENet (Lei & Shi, 2021), and MambaIR-v2 (Guo et al., 2025). Among these models, SRCNN (Dong et al., 2014) is a pioneering work that adopted CNNs to deal with SISR tasks. MHAN (Zhang et al., 2020) and HSENet (Lei & Shi, 2021) are two representative models specifically designed for RSISR tasks. MambaIR-v2 (Guo et al., 2025) is a latest model with

---

**Algorithm 1 Channel Shifting**

---

**Input:** $\mathbf{x}_t \in \mathbb{R}^{C \times H \times W}, \quad \mathbf{s} \in \mathbb{R}^{2C}$
**Output:** $\mathbf{x}_t^* = \text{CS}(\mathbf{x}_t, \mathbf{s}) \in \mathbb{R}^{C \times H \times W}$
  1: $\mathbf{g} \leftarrow \text{createGrid}(H, W)$          $/^*1 \times H \times W \times 2^*/$
  2: $\mathbf{s} \leftarrow \text{reshape}(\mathbf{s}, (C, 1, 1, 2))$
  3: $\widetilde{\mathbf{g}} \leftarrow \mathbf{g} + \mathbf{s}$             $/^*\text{Broadcast Addition}^*/$
  4: $\widetilde{\mathbf{g}}[...,0] \leftarrow 2 \times \frac{\widetilde{\mathbf{g}}[...,0]}{W-1} - 1$    $/^*\text{Normalize } x^*/$
  5: $\widetilde{\mathbf{g}}[...,1] \leftarrow 2 \times \frac{\widetilde{\mathbf{g}}[...,1]}{H-1} - 1$    $/^*\text{Normalize } y^*/$
  6: $\hat{\mathbf{g}} \leftarrow \text{reshape}(\widetilde{\mathbf{g}}, (C, H, W, 2))$
  7: $\hat{\mathbf{x}}_{\mathbf{t}} \leftarrow \text{reshape}(\mathbf{x}_t, (C, 1, H, W))$
  8: $\widetilde{\mathbf{x}}_{\mathbf{t}} \leftarrow \text{gridSample}(\hat{\mathbf{x}}_{\mathbf{t}}, \hat{\mathbf{g}}; \text{mode=``bilinear''})$
  9: $\mathbf{x}_t^* \leftarrow \text{reshape}(\widetilde{\mathbf{x}}_{\mathbf{t}}, (C, H, W))$
10: **return** $\mathbf{x}_t^*$

---

superior performance for image restoration. The additional quantitative results are provided in Table 6 of this supplementary material.

As illustrated in Table 6, it can be observed that although SRCNN (Dong et al., 2014) is highly efficient compared to other models, its performance is severely unsatisfactory. On the other hand, MHAN (Zhang et al., 2020) and HSENet (Lei & Shi, 2021) show better results, but their parameter counts and computational costs are excessively high, placing them beyond the scope of lightweight models. The model closest to our SFSN in SR performance is MambaIR-v2 (Guo et al., 2025). However, our SFSN still achieves more superior SR results with lower computational overhead, striking a better performance-efficiency equilibrium than MambaIR-v2 (Guo et al., 2025).

**Additional Visual Results**: To complement the qualitative results and provide more visual comparison, we exhibit extra visual results in Fig. 12, Fig. 13, Fig. 14, Fig. 15. It is observable that our method achieves superior recovery of HF information neglected by other models, which can induce blurring or artifacts within complex regions.

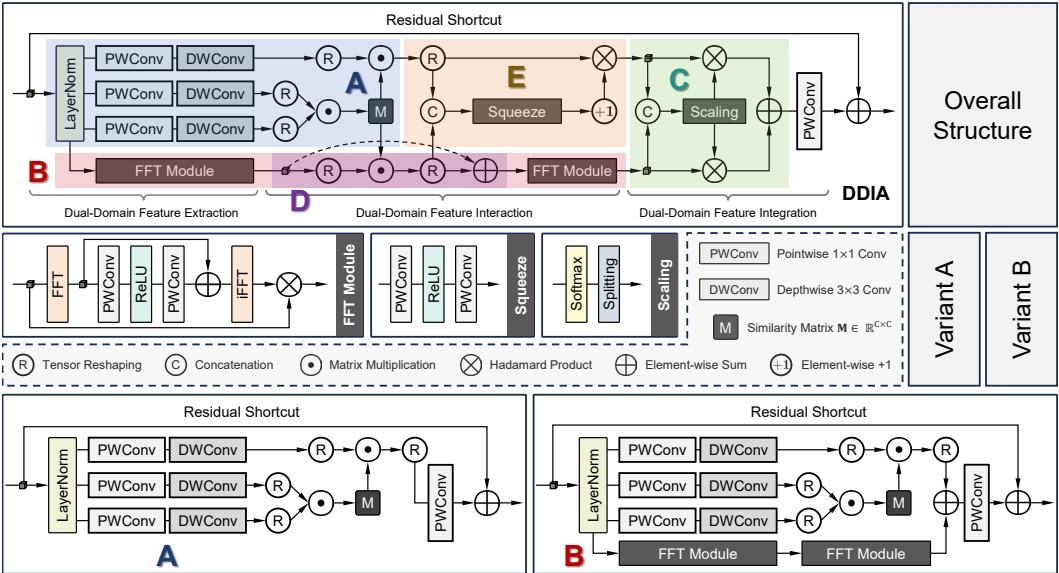

Figure 9: The variants for DDIA ablation study. We mark the main structural components of our DDIA with different colors. The variant **A** can be viewed as a simple self-attention, and the variant **B** is built by adding component **B** (i.e., the FFT branch) to the variant **A**. Other variants tested in the ablation study of our DDIA can be found in Fig. 10.

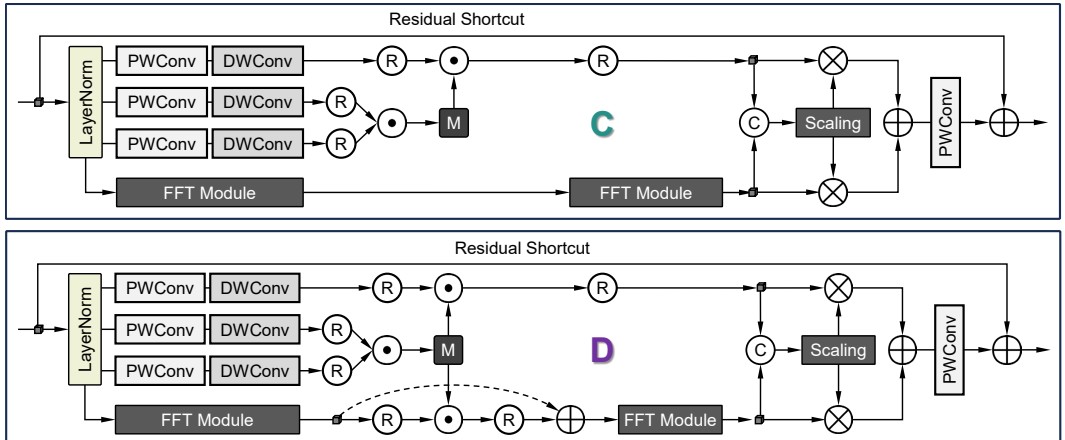

Figure 10: The variants **C** and **D** for DDIA ablation study. **C** is built by incorporating component **C** (i.e., feature scaling) into the variant **B**, and **D** is formed with component **D** (i.e., dual-domain feature interaction). When component **E** (squeeze) is adopted by the variant **D**, it then become the complete DDIA module.

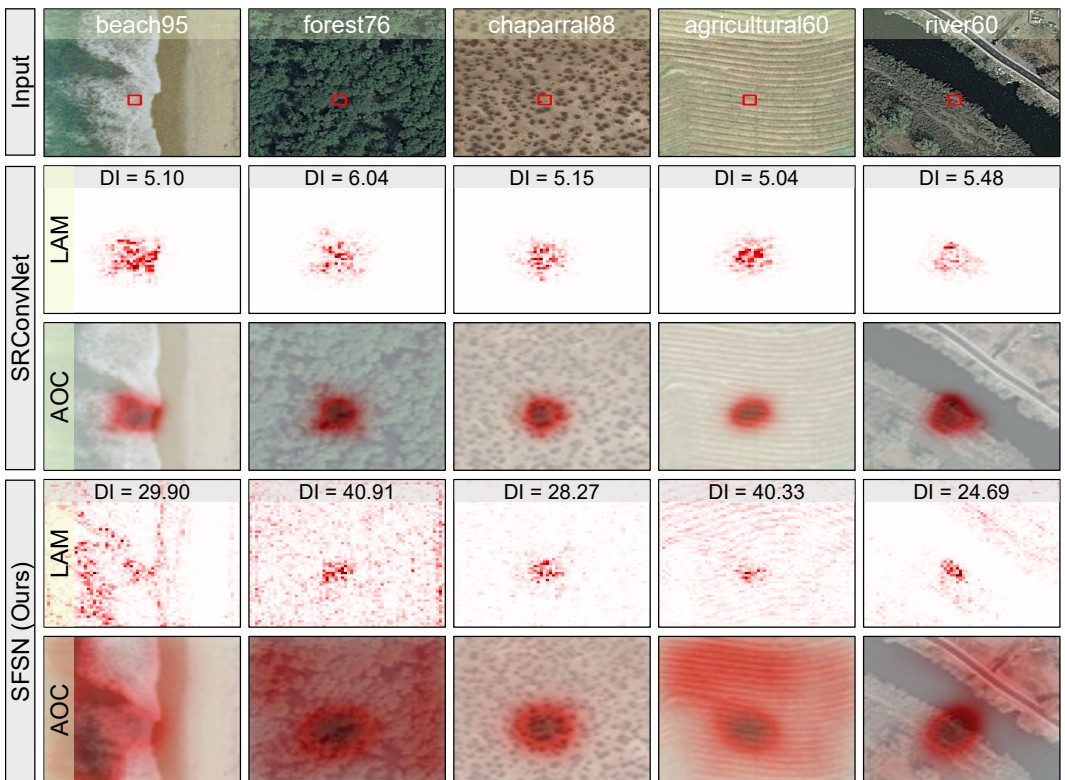

Figure 11: LAM and AOC comparison between SRConvNet (Li et al., 2025b) and the proposed SFSN on five testing images from UCMerced (Yang & Newsam, 2010).

Table 6: Quantitative results of advanced SISR models. SFSN+ represents the enhanced version of our model with geometric self-ensemble (Lim et al., 2017). The best and second-best results are marked in red and blue, respectively.

| Scales | SISR Models | Annual | Params [K] | FLOPs [G] | UCMerced PSNR | UCMerced SSIM | RSSCN7 PSNR | RSSCN7 SSIM | AID PSNR | AID SSIM |
|---|---|---|---|---|---|---|---|---|---|---|
| SR×2 | SRCNN | ECCV2014 | 57 | 52.7 | 32.94 | 0.9170 | 29.83 | 0.7954 | 34.65 | 0.9290 |
| | MHAN | TGRS2020 | 11203 | 2315.6 | 33.92 | 0.9283 | 30.06 | 0.8036 | 35.56 | 0.9390 |
| | HSENet | TGRS2022 | 5286 | 939.6 | 34.22 | 0.9327 | 30.15 | 0.8070 | 35.50 | 0.9383 |
| | MambaIRv2-L | CVPR2025 | 774 | 286.3 | 34.54 | 0.9345 | 30.25 | 0.8104 | 35.54 | 0.9386 |
| | SFSN (Ours) | — | 723 | 163.0 | 34.57 | 0.9350 | 30.26 | 0.8113 | 35.56 | 0.9397 |
| | SFSN+ (Ours) | — | 723 | 163.0 | 34.75 | 0.9365 | 30.30 | 0.8123 | 35.67 | 0.9401 |
| SR×3 | SRCNN | ECCV2014 | 57 | 52.7 | 28.91 | 0.8132 | 27.77 | 0.6936 | 30.55 | 0.8372 |
| | MHAN | TGRS2020 | 11287.5 | 1177.7 | 29.94 | 0.8391 | 28.00 | 0.7045 | 31.55 | 0.8603 |
| | HSENet | TGRS2022 | 5470 | 430.5 | 30.04 | 0.8433 | 28.02 | 0.7067 | 31.49 | 0.8588 |
| | MambaIRv2-L | CVPR2025 | 781 | 126.7 | 30.24 | 0.8491 | 28.08 | 0.7083 | 31.58 | 0.8607 |
| | SFSN (Ours) | — | 729 | 71.9 | 30.29 | 0.8519 | 28.10 | 0.7102 | 31.60 | 0.8622 |
| | SFSN+ (Ours) | — | 729 | 71.9 | 30.44 | 0.8550 | 28.14 | 0.7115 | 31.67 | 0.8628 |
| SR×4 | SRCNN | ECCV2014 | 57 | 52.7 | 26.92 | 0.7286 | 26.64 | 0.6278 | 28.48 | 0.7565 |
| | MHAN | TGRS2020 | 11351 | 711.9 | 27.63 | 0.7581 | 26.79 | 0.6360 | 29.39 | 0.7892 |
| | HSENet | TGRS2022 | 5433 | 270.1 | 27.75 | 0.7611 | 26.82 | 0.6378 | 29.32 | 0.7867 |
| | MambaIRv2-L | CVPR2025 | 790 | 49.6 | 27.88 | 0.7694 | 26.86 | 0.6408 | 29.40 | 0.7894 |
| | SFSN (Ours) | — | 738 | 41.6 | 27.89 | 0.7699 | 26.87 | 0.6419 | 29.46 | 0.7913 |
| | SFSN+ (Ours) | — | 738 | 41.6 | 28.09 | 0.7747 | 26.92 | 0.6437 | 29.51 | 0.7923 |

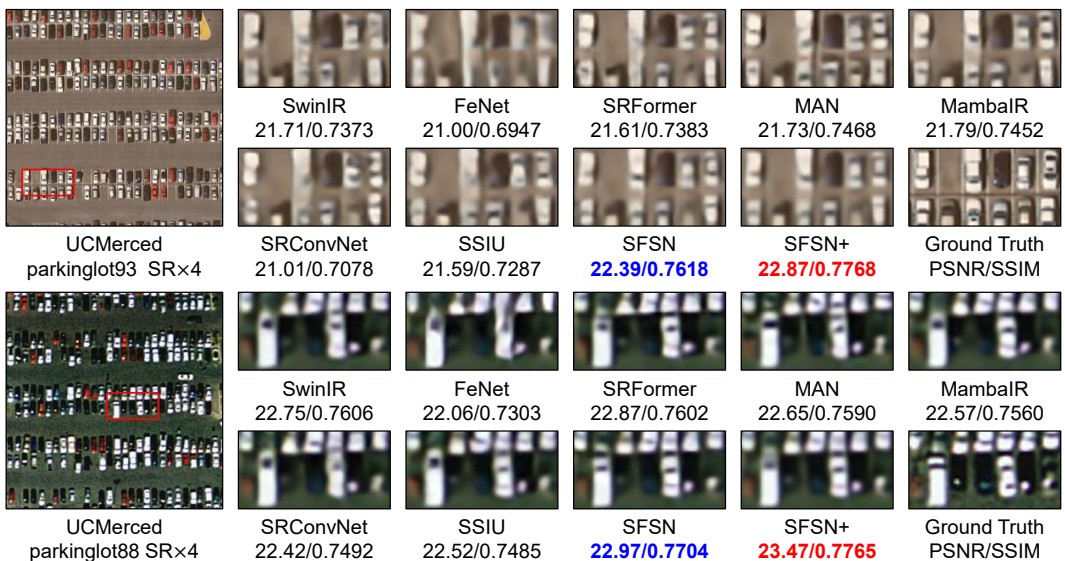

Figure 12: Visual results of compared RSISR models on two testing images from UCMerced (Yang & Newsam, 2010). The best and second-best results are marked with red and blue, respectively.

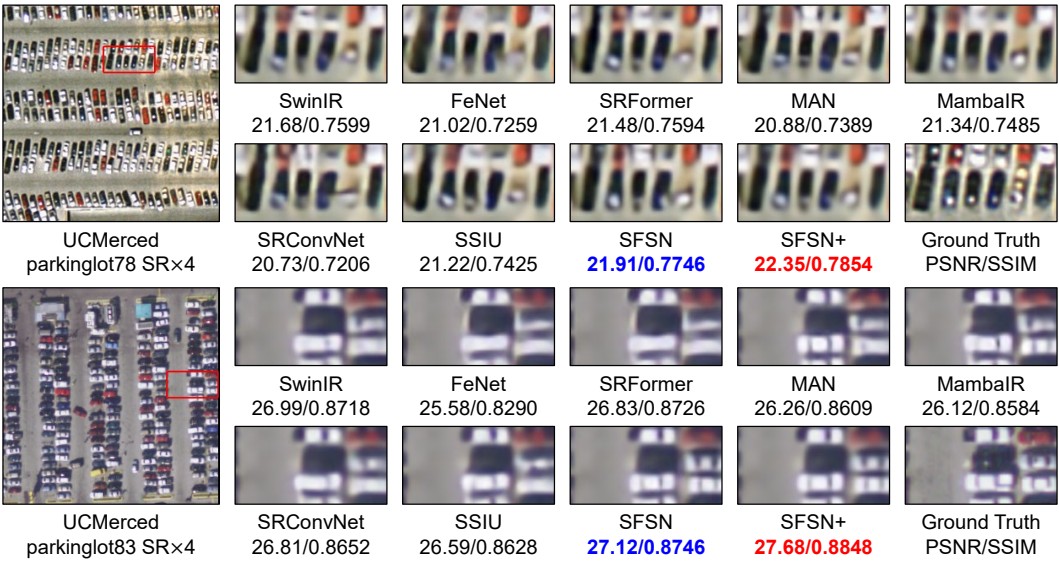

Figure 13: Visual results of compared RSISR models on two testing images from UCMerced (Yang & Newsam, 2010). The best and second-best results are marked with red and blue, respectively.

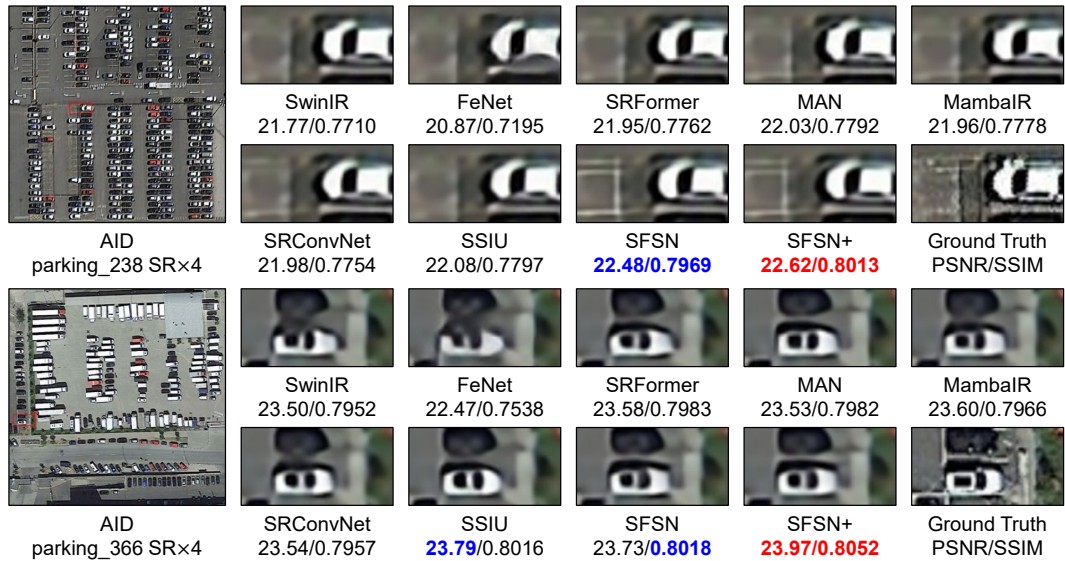

Figure 14: Visual results of several compared RSISR models on two testing images from AID (Xia et al., 2017). The best and second-best results are marked with red and blue, respectively.

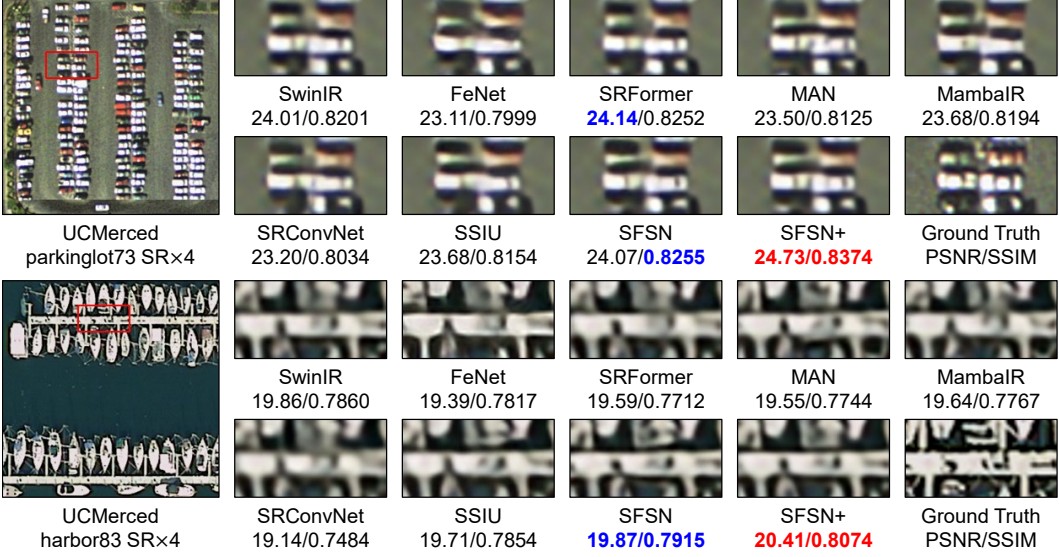

Figure 15: Visual results of compared RSISR models on two testing images from UCMerced (Yang & Newsam, 2010). The best and second-best results are marked with red and blue, respectively.

