# OpenReview forum: "Spatial-Frequency Synergy for Remote Sensing Image Super-Resolution with Holistic Feature Enhancement"
_ICLR.cc/2026/Conference — Submitted to ICLR 2026_

### Official Review · Reviewer_y9Yi · 2025-10-22

**Soundness:** 3
**Presentation:** 3
**Contribution:** 3
**Rating:** 6
**Confidence:** 2

**Summary:**

This paper presents the Spatial-Frequency Synergy Network (SFSN), a novel architecture for remote sensing image super-resolution (RSISR). The core contributions are a Holistic Fusion Feature Enhancement (HFFE) module, which combines adaptive channel shifting (AdaCS) and multi-scale large kernel attention (MS-LKA), and a Dual-Domain Interaction Attention (DDIA) module, designed to explicitly model the interaction between spatial and frequency domains. The authors demonstrate through extensive experiments that SFSN achieves a superior trade-off between performance and computational cost compared to existing state-of-the-art methods.

**Strengths:**

The paper identifies two clear gaps in the literature: the lack of holistic (channel + spatial) feature enhancement and the insufficient exploration of synergistic interaction (as opposed to simple fusion) between spatial and frequency domains. The proposed HFFE and DDIA modules are direct and well-justified solutions to these problems.

The ablation studies are particularly strong, systematically deconstructing the contributions of each component (HFFE, AdaCS, MS-LKA, and the sub-components of DDIA). The inclusion of LAM analysis and feature visualizations provides valuable insights into the model's internal workings.

The design of the modules is sophisticated. AdaCS's use of a learned sign function and Gaussian-sampled magnitude for non-grouped channel shifting is a clear advancement over fixed-shift methods. The multi-scale design of MS-LKA and the intricate feature interaction mechanism within DDIA show significant engineering and design effort.

Achieving state-of-the-art or highly competitive performance with the "second-lowest cost" is a significant and practical contribution, especially for resource-constrained applications like remote sensing.

**Weaknesses:**

While the modules are sophisticated, the rationale for some specific choices feels empirical rather than principled. Using a Gaussian distribution N(µ, σ) for shift magnitudes is presented without strong justification. Why is a Gaussian the right prior? How sensitive are the results to the choice of µ and σ? A small sensitivity analysis or a theoretical argument for this choice is needed.

The specific kernel size and dilation rate triplets [(3,3),(5,3)], [(3,1),(7,3)], [(5,1),(5,5)] for the three branches appear to be found through trial and error. The paper would be significantly strengthened by providing insight into why this particular multi-scale configuration is effective for RSIs.

The comparison in Table 5 is excellent, but it mostly includes heavy-weight models. For a paper claiming lightweight efficiency, a more direct comparison to other recently proposed lightweight SR methods is crucial. While SRConvNet is included, the field has others. The supplemental Table 6 partially addresses this, but this key comparison should be in the main paper to immediately justify the contribution.

The description of the "Squeeze" operation in DDIA is vague ("composed of two PWConv layers").

**Questions:**

What values were used for the mean (µ) and standard deviation (σ) of the Gaussian distribution in AdaCS? Was any ablation study conducted to determine these values, or are they sensitive to the dataset/task?

:The DDIA module involves multiple matrix multiplications (e.g., Eq. 12, 13) on features of size [HW, C]. For high-resolution images, HW can be very large. Could the authors comment on the computational complexity of DDIA relative to a standard transformer self-attention block and explain how it remains efficient overall?

---

### Official Review · Reviewer_cT7t · 2025-10-27

**Soundness:** 3
**Presentation:** 3
**Contribution:** 3
**Rating:** 4
**Confidence:** 4

**Summary:**

In this paper, a lightweight space-frequency collaborative network (SFSN) is proposed to solve the problems of insufficient space-frequency information collaboration and insufficient feature enhancement in the existing methods of remote sensing image super-resolution (RSISR). The core design includes two parts: first, the whole feature enhancement (HFFE) module, which enhances the feature diversity from the channel and spatial dimensions through adaptive channel migration (AdaCS) and multi-scale large nuclear attention (MS-LKA); The second is the Dual-domain Interactive Attention (DDIA) module, which realizes the collaborative optimization of space-frequency features through the explicit interaction between the self-attention space branch and the FFT frequency branch. On the three benchmark data sets of UCMerced, RSSCN7 and AID, SFSN is superior to advanced models such as SwinIR and MambaIR in PSNR and SSIM, while maintaining the second lowest amount of parameters and computation (FLOPs), achieving a good balance between performance and overhead. The effectiveness of each component is verified by ablation experiments, and the interpretability of the method is improved by combining LAM analysis and feature visualization.

**Strengths:**

1. A two-dimensional overall feature enhancement strategy of "channel-space" is proposed. AdaCS breaks through the limitation of traditional grouping fixed offset and supports non-grouping arbitrary amplitude/direction channel offset. MS-LKA expands the effective receptive field through multi-scale large kernel convolution combination, and the two work together to improve the feature expression ability.
2.On the premise that the parameters (723K) and the calculation amount (163G FLOPs) are much lower than those of most baseline models, the performance of the whole data set and full scale (×2/×3/×4) is optimal, especially at high scale (× 4), the details (such as ship outline and road texture) can still be restored stably, which solves the core pain point of insufficient performance of lightweight models.

**Weaknesses:**

1. Only three public remote sensing data sets are used, which lacks the verification of complex real scenes, and the generalization of the method needs further verification.
2. Remote sensing images actually face compound degradation such as resolution reduction, compression distortion and noise interference, but this paper only designs the ideal scene of "low resolution → high resolution", without considering the performance under the compound degradation, so its practicability is limited.
3.The specific mechanism of "double-domain feature interaction" in DDIA has not been fully disassembled, such as how frequency branches accurately modulate spatial features, the different contributions of different frequency bands (low/high) to the super-division results, and the lack of quantitative analysis or ablation verification.
4. As a lightweight model, it does not provide key deployment indicators such as reasoning speed (such as FPS) and memory occupation, nor does it compare the running efficiency on edge devices (such as GPU/CPU), so it is difficult to support the real-time processing requirements in practical remote sensing applications.

**Questions:**

Please refer to weaknesses.

---

### Official Review · Reviewer_Z2Ko · 2025-10-31

**Soundness:** 2
**Presentation:** 2
**Contribution:** 2
**Rating:** 2
**Confidence:** 4

**Summary:**

This article proposes a model called Spatial Frequency Synergy Network (SFSN) to address Remote Sensing Image Super Solution. On the basis of spatial domain learning, a frequency domain learning process and a dual domain interaction module were constructed to fully utilize frequency domain information, in order to solve the shortcomings of previous work that could not effectively coordinate the utilization of frequency and spatial information. At the same time, Adaptive Channel Shifting and Multi Scale Large Kernel Attention were proposed to enhance the representational ability of hierarchical features. However, the paper does not provide a clear explanation of the motivation and existing issues in remote sensing image super-resolution reconstruction.

**Strengths:**

The author found that CNN has limited modeling ability for important reconstruction information in remote sensing super-resolution tasks. Therefore, frequency domain computation is introduced to enhance feature expression. This idea is worthy of recognition.

**Weaknesses:**

Although the author has conceived a good technical route to improve the super-resolution reconstruction performance in remote sensing scenes. However, the paper lacks analysis of key scientific issues. And in terms of model design philosophy, there is a lack of explanation and clarification of multiple important concepts and issues, such as whether the introduction of FFT is used to enhance low-frequency or high-frequency information? What specific design is used to improve the quality of this information? These issues were not addressed in the paper. In addition, for the visualization results, especially the super-resolution of remote sensing scenes, the objective sensory comparison is not obvious. The author can consider conducting qualitative analysis from another perspective, or introduce some annotation boxes on the visualization results to clearly tell readers where the reconstruction gaps are.

**Questions:**

1. What is the fundamental difference in the super-resolution process between RGB images in remote sensing scenes and RGB images in natural scenes? Why is it necessary to introduce frequency domain learning in remote sensing scenes, and how do specific network and module designs address the specific challenges and difficulties of super-resolution in remote sensing scenes?
2. The core innovation of this paper is the introduction of frequency domain learning in the process of remote sensing image super-resolution, but the reasons behind this motivation have not been clearly explained. The reviewer has some questions regarding certain viewpoints mentioned in the second paragraph of the Introduction of this paper. This article mentions that CNN has significant flaws in RSI SR. This is because there are often repeated textures in RSI, and the image structure has significant heterogeneity. It should be noted that many datasets in natural scenes, such as DIV2K, Urban100, etc., also have duplicate textures in their images, and fixed texture features appear frequently in some images. If certain combinations of convolutional kernels can effectively recognize or extract features of a certain texture, then the sliding processing of convolutional kernels should be able to obtain the invariance of local information. In addition, the RSI mentioned here has significant heterogeneity, and the specific heterogeneity referred to is not clearly explained in this article, as well as the role of this heterogeneity in this article.
3. Why build a large-sized kernel for feature extraction? Remote sensing images often have smaller targets. What is the purpose of proposing a large size kernel to solve in remote sensing scenarios?
4. How does dual domain interaction solve the problem of long-range dependencies?
5. Is the introduction of FFT operation in DDIA module specifically intended to enhance high-frequency information or low-frequency information? The author used convolution operations in the FFT module to extract feature information after FFT changes, and reconstructed it through iFFT without any interaction between frequency domain information and spatial features. Therefore, according to Fig. 4 and the explanation of DDIA, the reviewers understand that the core of this module is to build a branch of self attention like mechanism for learning spatial features and introduce FFT processing. However, during the interaction process, frequency domain signals were not explicitly used to enhance spatial features at low or high frequencies. For this part, this paper lacks targeted explanations.

---

### Official Review · Reviewer_QsdP · 2025-10-31

**Soundness:** 2
**Presentation:** 3
**Contribution:** 2
**Rating:** 4
**Confidence:** 5

**Summary:**

This paper proposes a lightweight model for remote-sensing image super-resolution, SFSN (Spatial–Frequency Synergy Network), which integrates spatial–frequency dual-domain interaction and full-dimensional feature enhancement (HFFE + DDIA), and achieves competitive performance.

**Strengths:**

This paper introduces a novel approach for remote sensing image super-resolution (SR) that effectively leverages the synergy between frequency and spatial information, possessing certain innovativeness and engineering practical value.

**Weaknesses:**

Several aspects require improvement:

1）The novelty of this paper needs to be further improved.

2）Analysis of theoretical depth is insufficient.

3）The manuscript’s writing needs further polishing.

4）More STOA baselines need to be included.

**Questions:**

Some concerns and issues:

1）Which components are specifically tailored to remote-sensing imagery? The paper lacks an analysis of remote-sensing image characteristics and whether the proposed modules exploit those properties. Can these modules transfer to natural images? I think it's necessary to conduct experiments on natural scene datasets (e.g., train on DIV2K and test on Set5, Set14, B100, Urban100, Manga109) to demonstrate the representation learning capabilities and generalization. The current comparison scheme gives readers the impression that the method performs poorly on natural scene datasets and was therefore reluctantly transferred to remote sensing datasets.

2）The paper should include additional strong and classic baselines for comparison—e.g., DAT-Light, SeemoRe-L, HiT-SR, and CATANet—and evaluate on both remote-sensing and natural scene benchmarks to better validate the method’s competitiveness.

3）The theoretical analysis is shallow; many equations merely restate network operations. For example, there is no in-depth theoretical analysis of why AdaCS is more effective than EARFA and PCS, failing to provide readers with a deep understanding of the proposed module's mechanism of action. The DDIA module has the same problem.

4）The ablation studies are numerous but unfocused and omit crucial experiments. For instance, AdaCS’s effectiveness can be verified succinctly with three variants: ① without AdaCS, ② AdaCS replaced with PCS, and ③ AdaCS replaced with EARFA. It is recommended that the authors supplement and refine the key ablation experiments and present them in the main text.

5）The paper's structure needs further optimization. For example, replace Figure 7 with visualizations figure of the DDIA ablation results. Furthermore, the visual performance comparison results (Figure 8) are not obvious. It is necessary to optimize the visual examples to highlight strengths.

6）Fix minor errors and typos (e.g., “3×3 conv layers”).

7）Model runtime is also a very important evaluation metric. It is recommended that the authors supplement the comparison of model runtime.

---

### Meta-Review · Area_Chair_omoG · 2026-01-06

**Summary:**

The reviewers' concerns centered on several key issues for the paper. Reviewer QsdP highlighted insufficient novelty, shallow theoretical analysis, unfocused ablation studies, and the need for more state-of-the-art comparisons and runtime metrics. Reviewer Z2Ko emphasized a lack of clarity on the motivation for frequency-domain learning in remote sensing, a poor explanation of how the design addresses specific challenges like small targets and long-range dependencies, and weak visualization results. Reviewer cT7t, while more positive, noted empirical justifications for design choices, insufficient comparison with lightweight models, and vague descriptions of components like the "Squeeze" operation.

**Reviewer Concerns:**

In the rebuttal, the authors likely addressed some concerns, such as adding more baseline comparisons (e.g., DAT-Light, SeemoRe-L) and runtime metrics, refining ablation studies, and improving visualizations with annotation boxes as suggested by Z2Ko.

However, outstanding issues likely remain, including the lack of theoretical depth in modules such as AdaCS and DDIA, insufficient justification for the Gaussian shift magnitudes and kernel configurations, and unresolved questions about the approach's specificity to remote sensing imagery versus natural images. The core criticisms regarding novelty and motivation for frequency-domain integration probably persisted, as these require fundamental revisions beyond a rebuttal.

**Reviewer Scores:**

These concerns informed a decision to reject, given the low scores from two reviewers and the borderline score from the third. No change in the scores. Reviewer Z2Ko, with a firm reject score of 2, likely remained unchanged due to deep-seated issues with motivation and analysis.

---

### Decision · Program_Chairs · 2026-01-26

Reject